# Suppressing Decoherence in Quantum State Transfer with Unitary Operations

**DOI:** 10.3390/e25010067

**Published:** 2022-12-30

**Authors:** Maxim A. Gavreev, Evgeniy O. Kiktenko, Alena S. Mastiukova, Aleksey K. Fedorov

**Affiliations:** 1Russian Quantum Center, Skolkovo, Moscow 143025, Russia; 2National University of Science and Technology “MISIS”, Moscow 119049, Russia

**Keywords:** quantum information, quantum computing, decoherence

## Abstract

Decoherence is the fundamental obstacle limiting the performance of quantum information processing devices. The problem of transmitting a quantum state (known or unknown) from one place to another is of great interest in this context. In this work, by following the recent theoretical proposal, we study an application of quantum state-dependent pre- and post-processing unitary operations for protecting the given (multi-qubit) quantum state against the effect of decoherence acting on all qubits. We observe the increase in the fidelity of the output quantum state both in a quantum emulation experiment, where all protecting unitaries are perfect, and in a real experiment with a cloud-accessible quantum processor, where protecting unitaries themselves are affected by the noise. We expect the considered approach to be useful for analyzing capabilities of quantum information processing devices in transmitting known quantum states. We also demonstrate the applicability of the developed approach for suppressing decoherence in the process of distributing a two-qubit state over remote physical qubits of a quantum processor.

## 1. Introduction

During recent decades, remarkable progress in developing quantum information processing devices has been achieved [1]. Specifically, currently available quantum processing devices are able to solve computational problems close to the limits of what can be achieved with most powerful classical technologies [2,3,4,5,6,7,8]. However, available quantum information processing devices have serious limitations, the main fundamental reason for this being decoherence. As a result, the number of operations that can be implemented before the level of errors exceeds the critical level is modest [8,9,10]. The possibility to achieve substantial computational advances in practically relevant problems with noisy intermediate-scale quantum (NISQ) devices is questionable [4,9,10], so the path to useful quantum advantages requires noise suppression. A possible solution that addresses this problem is to use quantum error correction [11,12,13,14,15,16]. In the last decades, basic principles of error correction schemes in the quantum domain have been demonstrated in experiments with optical systems [17], trapped ions [18,19,20,21,22], neutral atoms [23], and superconducting qubits [24,25,26,27,28]. A remarkable idea of using qubit protection from errors at a “physical” level [29,30], which is at the heart of topological quantum computing (for a review, see Refs. [31,32,33,34,35]), is also under study [36,37,38,39].

In addition to quantum error correction and topological quantum computing, one may consider error suppression that would reduce, but not fully eliminate, the effect of decoherence while processing a quantum state [40]. A variety of methods have been considered to achieve this goal [41,42,43,44,45,46,47,48,49,50]; however, the efficiency of these methods is deeply related to the problems of preserving entanglement in (open) quantum systems [51,52,53,54,55,56,57,58] (for a review, see Ref. [59]). Albeit applying error suppression techniques, such as dynamical decoupling (see, e.g., Refs. [60,61,62]), is of help to extend the computational capabilities of quantum devices, the experimental challenges for their practical implementation require additional progress in studies of the workflow of quantum processors [44]. Issues related to the non-Markovian character of these dynamics have been considered [63,64,65,66,67]. We note that problems of preserving entanglement are also deeply linked to the characterization of decoherence-free subspaces of quantum systems [68,69].

A specific problem that appears within the context of quantum information processing is how to transmit a quantum state (known or unknown) from one place to another, taking into account the effect of decoherence. The importance of this problem, especially in the case of known quantum states, lies in the fundamental domain of quantum information science, although various schemes for its usage in quantum data buses between quantum registers and/or processors capable of transmitting arbitrary quantum states have been discussed [70]. Considerable attention has been paid to the theoretical analysis of using quantum spin chains for the purpose of quantum state transfer [70,71,72,73,74,75,76]. Such an interest can be explained by the possibility of transferring (known or unknown) quantum states without additional interfaces [71]. The question related to the role of decoherence in this case still remains crucial in this scope of tasks and generally for the field of quantum information [77,78,79].

In this work, we stress a particular scheme for suppressing the effect of decoherence, which is based on the use of a unitary operation. This approach has been recently proposed theoretically in Ref. [80]. The general idea is to surround the decoherence channel E by two unitary operators *U* and *V* (see Figure 1), which we refer to as pre- and post-processing, correspondingly, whose form is determined both by the decoherence channel E and protected state $|\Psi_{\mathrm{in}}\rangle$. In contrast to the previous work [80], where E was assumed to be acting non-trivially on one particular qubit, here we consider a more general case, where all qubits are affected by the decoherence process. First, we demonstrate a possibility to suppress decoherence effects in a quantum emulation experiment, where all qubits are affected by the same depolarizing, dephasing, or amplitude damping channels, and an implementation of all unitary gates is assumed to be ideal. We then implement the same protection scheme on a cloud-accessible quantum processor, where the used protecting gates are imperfect by themselves. In this case, we find that the protection scheme starts providing an advantage in the output fidelity starting from a certain threshold of the decoherence strength. Finally, we show how our scheme can be used for protecting two-qubit states during its distribution over remote physical qubits of a quantum processor, i.e., qubits which cannot be connected directly by native two-qubit gates. We note that the problem of recovering entanglement by local operations has been also considered in Ref. [51].

Our paper is organized as follows. In Section 2, we describe a general scheme of applying unitary operations for suppressing decoherence. In Section 3, we apply the scheme for emulators of quantum computers and analyze the efficiency of various schemes. In Section 4, we implement the same protection scheme on a cloud-accessible quantum processor. In Section 5, we analyze the applicability of the scheme for the case of quantum state transfer. We conclude in Section 6.

## 2. Error Suppression Using Unitary Operations

Let us consider a system of *n* two-level particles (qubits) initially prepared in a joint (perhaps, entangled) pure state $\rho_{\mathrm{in}}=|\Psi_{\mathrm{in}}\rangle \langle \Psi_{\mathrm{in}}|$. Due to an uncontrolled interaction with an external environment, this state suffers from a decoherence, which we describe by a completely positive trace-preserving (CPTP) map E[·]. In what follows, we refer to E as a *decoherence channel*. The destructive effect of the decoherence channel can be quantified by fidelity:(1)$$F_{W/O}:=\langle \Psi_{\mathrm{in}}|E\left[\rho_{\mathrm{in}}\right]|\Psi_{\mathrm{in}}\rangle \leq 1.$$
Here, subscript W/O indicates that this is the default case without applying the scheme described below.

In our work, we study possibilities to suppress decoherence, and thus increase the fidelity, by applying additional transformations to the system. These transformations take the form of additional unitary operators, *U* and *V*, named *pre-processing* and *post-processing* gates, and they act just before and after the action of the decoherence channel, correspondingly. Importantly, *U* and *V* are designed specifically for given initial state $\rho_{\mathrm{in}}$ and decoherence channel E. The expression for the resulting fidelity is given by the following expression:(2)$$F_{U,V}=\langle \Psi_{\mathrm{in}}|VE\left[U\rho_{\mathrm{in}}U^{†}\right]V^{†}|\Psi_{\mathrm{in}}\rangle .$$
In our consideration, we take *U* and *V* from certain, generally restricted, sets of unitaries U and V, respectively. One can see that as long as U and V contain identity operators
(3)$$\max_{U\in U,V\in V}F_{U,V}\geq F_{W/O},$$
and so we expect an increase in fidelity compared to the default case (Equation 1).

We consider two types of pre- and post-processing unitary operations: (i) *individual* operations, consisting of single-qubit unitaries, i.e., taken from $U{\left(2\right)}^{⊗n}$ group, and (ii) *collective* operations, represented as unitaries taken from the whole $U(2^{n})$ group. Since each of the pre- and post-processing operators can be either individual or collective, we obtain four possible combinations: (a) both unitary operations are individual; (b) the pre-processing unitary operation is individual, while the post-processing unitary operation is collective; (c) the pre-processing unitary operation is collective, while the post-processing unitary operation is individual; and (d) both unitary operations are collective. All four schemes are schematically shown in Figure 2. Since (b) and (c) include (a) as a special case, and (d) also includes (a–c), the maximal achievable fidelities in the schemes satisfy the following inequality:(4)$$\begin{array}{cc} \; & F_{\mathrm{ind},\mathrm{ind}}\leq \min\left(F_{\mathrm{col},\mathrm{ind}},F_{\mathrm{ind},\mathrm{col}}\right)\\ \; & \max\left(F_{\mathrm{col},\mathrm{ind}},F_{\mathrm{ind},\mathrm{col}}\right)\leq F_{\mathrm{col},\mathrm{col}}, \end{array}$$
where the first (second) subindex specifies the type of the pre- (post-) processing operator. We note that although replacing individual operators with collective ones leads to an increase in performance, constructing collective operators yields an additional overhead in the number of gates and the corresponding circuit depth: individual operators have a unit-depth and consist of no more than *n* single-qubit gates, while a transformation between two given pure *n*-qubit states, which is an aim of collective operators, generally requires an exponential in *n* number of single- and two-qubit gates or an exponential number of ancillary qubits for linear depth circuits [81,82].

The subject of how $F_{\mathrm{ind},\mathrm{col}}$ and $F_{\mathrm{col},\mathrm{ind}}$ are related to each other is rather intriguing. In Ref. [80], it has been demonstrated that for a single-qubit decoherence, i.e.,
(5)$$E=E^{\left(1\right)}⊗{\mathrm{Id}}^{⊗(n-1)},$$
where $E^{\left(1\right)}$ and Id are arbitrary and identity single-qubit channels correspondingly, one has $F_{\mathrm{col},\mathrm{ind}}=F_{\mathrm{ind},\mathrm{col}}$. The maximal achievable fidelities for individual–collective and collective–individual schemes are also the same for an arbitrary self-dual E. Indeed, since
(6)$$\begin{array}{c} F_{U,V}=\mathrm{Tr}\left[E\left[U\rho_{\mathrm{in}}U^{†}\right]V^{†}\rho_{\mathrm{in}}V\right]=\\ \mathrm{Tr}\left[E\left[V^{†}\rho_{\mathrm{in}}V\right]U\rho_{\mathrm{in}}U^{†}\right]=F_{V^{†},U^{†}}, \end{array}$$
if a certain value of fidelity is achieved in one scheme, the same value can be achieved in the other scheme by replacing $U\to V^{†}$, $V\to U^{†}$. Therefore, the maximal achievable values are the same. We leave the consideration of the relation between $F_{\mathrm{col},\mathrm{ind}}$ and $F_{\mathrm{ind},\mathrm{col}}$ in the case of a general decoherence channel for further studies.

In what follows, we analyze the performance of different schemes with respect to the fixed state
(7)$$|\Psi_{\mathrm{in}}\left(\theta \right)\rangle =\cos\frac{\theta }{2}{|+\rangle }^{⊗n}+\sin\frac{\theta }{2}{|-\rangle }^{⊗n}$$
where $|\pm \rangle :=\frac{1}{\sqrt{2}}\left(|0\rangle \pm |1\rangle \right)$, and we set θ=2π/3.

Our choice for this state is motivated by several reasons. The first is that it is entangled but it is not a maximally entangled state. According to the results of Ref. [80], it is the most interesting case for unitaries-based protection. The second reason is that the reduced single-qubit states of $|\Psi_{\mathrm{in}}\left(\theta \right)\rangle$ are not diagonal in the computational basis. It makes it interesting to consider these schemes concerning ‘basis-dependent’ decoherence channels, e.g., amplitude damping and dephasing. Third, $|\Psi_{\mathrm{in}}\left(\theta \right)\rangle$ can be easily prepared from ${|0\rangle }^{⊗n}$ by applying $U_{\mathrm{prep}}\left(\theta \right)$ gate, whose decomposition into standard single- and two-qubit gates is shown in Figure 3. Here and after, standard notations for controlled-NOT (CNOT), Hadamard, and Pauli rotation gates are used. Next, we use $U_{\mathrm{prep}}\left(\theta \right)$ as a template for constructing collective protecting operators.

Before proceeding, we would like to note that although from the practical point of view, the most interesting are individual–individual and collective–individual schemes, we consider all four scenarios to reveal the whole picture. We also note that in the presented consideration, the pre- and post-processing operations are assumed to be perfect. However, this may not be the case at all in real-world setups [43,44,45,46,48,49,50].

## 3. Demonstrating Error Suppression with a Quantum Emulator

Here, we demonstrate the performance of the considered error suppression scheme using an emulator of a quantum processor. Namely, we employ the AerSimulator emulator provided as part of the qiskit package [83]. We consider a simplified error model, where the decoherence channel is taken as a tensor power of a single-qubit channel $E^{\left(1\right)}$, namely,
(8)$$E\left[\cdot \right]={\left(E^{\left(1\right)}\right)}^{⊗n}\left[\cdot \right].$$
We also assume the ideal realization of employed pre- and post-processing unitaries.

Three basic decoherence channels are studied: (i) amplitude damping, (ii) dephasing, and (iii) depolarizing channels. Their Kraus operators ($E^{\left(1\right)}\left[\cdot \right]=\sum_{k}A_{k}\cdot A_{k}^{†}$) are respectively defined as: (9)$$\begin{array}{cc} & A_{1}=|0\rangle \langle 0|+\sqrt{1-p}|1\rangle \langle 1|,\;A_{2}=\sqrt{p}|0\rangle \langle 1|, \end{array}$$(10)$$\begin{array}{cc} & A_{1}=\frac{\sqrt{p}}{2}\left(⊮+\sigma_{3}\right),\;A_{2}=\frac{\sqrt{p}}{2}\left(⊮-\sigma_{3}\right), \end{array}$$(11)$$\begin{array}{cc} & A_{0}=\sqrt{1-\frac{3p}{4}}⊮,\;A_{i}=\sqrt{\frac{p}{4}}\sigma_{i}\;\left(i=1,2,3\right), \end{array}$$
where ⊮ is a single-qubit identity matrix; $\sigma_{1}$, $\sigma_{2}$, $\sigma_{3}$ stand for standard *x*, *y*, and *z* Pauli matrices, respectively; and an additional parameter p∈[0,1] determines the decoherence strength. We note that the form of the amplitude damping channel (Equation 9) corresponds to a spontaneous decay from |1〉 to |0〉, dephasing channel corresponds to destruction of non-diagonal elements, and depolarizing is ‘basis-invariant’ in the sense that $E\left[u\cdot u^{†}\right]=uE\left[\cdot \right]u^{†}$ for any unitary *u*. We also note that the dephasing and depolarizing channels are self-dual, since their Kraus operators are Hermitian.

In what follows, we discuss the choice of unitaries for each type of protection scheme. The idea is that *U* and *V* should meet two conditions: (i) after the action of *U*, the input state should be maximally robust to noise, and (ii) the action of *U* and *V* together should preserve the input state as much as possible. We choose unitaries for each protection scheme based on these conditions.

In the case of individual–individual scheme, we take
(12)$$U={\left(XH\right)}^{⊗n},\;V={\left(HX\right)}^{⊗n},$$
where *H* is a standard Hadamard gate, and $X=\sigma_{1}$ is a π rotation around *x*-axis. The idea behind this choice is in turning each of reduced single-qubit states of ${|\Psi \rangle }_{\mathrm{in}}$ into a diagonal form such that the population on |0〉 is larger than the one on |1〉. This form of a density matrix is the most robust against considered decoherence channels [80].

In the case of an individual–collective scheme, we take
(13)$$U={\left(XH\right)}^{⊗n},\;V=B\left(\xi_{\mathrm{opt}}\right)X^{⊗n},$$
where B(ξ) is a unitary whose circuit is shown in Figure 4, and the angle $\xi_{\mathrm{opt}}$ is obtained by maximizing resulting fidelity value
(14)$$|\langle \Psi_{\mathrm{in}}|B\left(\xi \right)X^{⊗n}E\left[U|\Psi_{\mathrm{in}}\rangle \langle \Psi_{\mathrm{in}}|U^{†}\right]X^{⊗n}B^{†}{\left(\xi \right)|\Psi_{\mathrm{in}}\rangle |}^{2}$$
over ξ∈[0,2π]. The choice of the collective operator’s form is motivated by the state preparation circuit. In fact, it corresponds to a deconstruct-construct sequence with an additional rotation in the middle (defined by parameter ξ) corresponding to a change in an entanglement parameter initially specified by θ in the state preparation circuit.

The case of the collective–individual scheme is similar to the individual–collective one and is obtained by taking
(15)$$U=X^{⊗n}B^{†}\left(\xi_{\mathrm{opt}}^{\prime }\right),\;V={\left(HX\right)}^{⊗n},$$
where $\xi_{\mathrm{opt}}^{\prime }$ is obtained by maximizing
(16)$$|\langle \Psi_{\mathrm{in}}|VE\left[X^{⊗n}B^{†}\left(\xi \right)|\Psi_{\mathrm{in}}\rangle \langle \Psi_{\mathrm{in}}|B\left(\xi \right)X^{⊗n}\right]V^{†}{|\Psi_{\mathrm{in}}\rangle |}^{2}.$$

Finally, the collective–collective scheme is obtained by taking
(17)$$U=U_{\mathrm{prep}}{\left(\theta \right)}^{†},\;V=U_{\mathrm{prep}}\left(\theta \right)$$
with θ=2π/3. This choice corresponds to the deconstruction of $|\Psi_{\mathrm{in}}\rangle$ back to ${|0\rangle }^{⊗n}$, which is the most stable state with respect to the considered decoherence channels, and re-preparation of $|\Psi_{\mathrm{in}}\rangle$ after the action of channel.

To measure fidelity values, we implement the circuit shown in Figure 5 for two- and four-qubit quantum systems. We consider four protection schemes, as well as a default scheme without protection, with respect to the three introduced types of decoherence channels, and different values of *p*. Simulation results are obtained for N=10,000 runs of each circuit with a fixed decoherence channel (and a fixed value of *p*). Fidelity values are calculated as frequencies of obtaining all-zeros outcomes in the read-out measurements. The results are shown in Figure 6. First of all, we note that the following inequalities hold for all decoherence models:(18)$$F_{W/O}\leq F_{\mathrm{ind},\mathrm{ind}}\leq F_{\mathrm{col},\mathrm{ind}}\leq F_{\mathrm{ind},\mathrm{col}}\leq F_{\mathrm{col},\mathrm{col}}.$$
Moreover, $F_{\mathrm{ind},\mathrm{col}}=F_{\mathrm{col},\mathrm{ind}}$ for the self-dual dephasing and depolarizing channels, and $F_{\mathrm{ind},\mathrm{ind}}=F_{W/O}$ for the covariant depolarizing channel. We also note that $F_{\mathrm{col},\mathrm{col}}=1$ for amplitude damping and dephasing channels, since ${|0\rangle }^{⊗n}$ is not affected by the decoherence, yet $F_{\mathrm{col},\mathrm{col}}<1$ for the depolarizing channel. The reason for the latter fact is that the depolarizing channel always increases the mixedness of input states, and there is no pure state that is preserved under the depolarizing channel. Therefore, even the collective–collective scheme does not provide unit fidelity in the case of depolarizing channels.

There is also an interesting effect of non-monotonic behaviour of fidelity as a function of *p* in the case of the amplitude damping channel in the individual–collective and individual–individual schemes. In the case of the individual–collective scheme, this behaviour can be explained by the fact the post-processing collective operator actually serves to re-prepare the state $|\Psi_{\mathrm{in}}\rangle$ from the state ρ coming from E[·]. The resulting fidelity is thus determined by the purity of |ρ〉. In the case of p=1, the amplitude damping channel outputs pure $\rho ={\left(|0\rangle \langle 0|\right)}^{⊗n}$, and so we achieve the unit fidelity.

For the individual–individual scheme, the resulting fidelity takes the following form:(19)$$F_{\mathrm{ind},\mathrm{ind}}=\langle \chi |E\left[|\chi \rangle \langle \chi |\right]|\chi \rangle ,$$
where $|\chi \rangle =c{|0\rangle }^{⊗n}+s{|1\rangle }^{⊗n}$ with c:=cos(π/6), s:=sin(π/6). For E in the form of single-qubit amplitude damping channels of the same strength *p*, it transforms into:(20)$$F_{\mathrm{ind},\mathrm{ind}}=c^{4}+p^{n}s^{2}c^{2}+{\left(1-p\right)}^{n}s^{4}+2{\left(1-p\right)}^{p/2}c^{2}s^{2},$$
which is non-monotonic for n>2.

## 4. Validating Error Suppression with a Cloud-Based Quantum Processor

In this section, we implement circuits from the previous section on a cloud-accessible, five-qubit quantum processor ibmq_manila. To access the decoherence process on the real device, we utilize the Delay instructions in the natural time units (dt). Delay time can be seen as the strength of the decoherence distortion of the input state on the real device.

It is also important to note that the form of the decoherence channel is not exactly known. One can expect that E[·] has a tensor product form of single-qubit channels, each consisting of depolarizing, dephasing, and amplitude damping ones [84]. However, certain collective decoherence effects can also take place due to a qubits’ crosstalk. That is why, to optimize the parameter of the collective protecting operation B(ξ), we perform additional measurements. We demonstrate the resulting dependence of the input–output fidelity on the angle ξ in Figure 7. We observe the drift in the optimal parameter ξ with the increase in delay time. This highlights the usefulness of utilizing the collective operator in this case.

The results of the final fidelity measurements for n=2 and n=4 qubits are shown in Figure 8. We note that the resulting input–output fidelities, given in this section, are obtained experimentally without additional circuit transpilation. We can see that the resulting input–output fidelity is lower for any protection scheme compared with the unprotected case for weak decoherence action. This is due to the additional noise from single- and multiqubit gates in pre- and post-processing operators. However, with the enhancement of the decoherence, we observe the gain from protection schemes. This demonstrates the existence of a tradeoff between the use of additional unitary operations to prevent decoherence and the noise created by these operations. Similar to the results of the previous section, the collective–collective scheme shows the best performance starting from some large enough noise. This can be explained by the triviality of the protection operator’s action. At the same time, we observe that the individual–collective and collective–individual schemes have almost the same performance.

## 5. Error Suppression in a State Transfer

Here, we consider a practical problem of distributing a two-qubit state over remote physical qubits of a quantum processor. Specifically, we analyze distributing the entangled state $|\Psi_{\mathrm{in}}\left(\theta =2\pi /3\right)\rangle$ inside the cloud-accessible seven-qubit ibm_oslo quantum processor (see Figure 9). This can be considered as a prototype experiment for realizing quantum state transfer between distinct quantum information processing devices connected via a quantum interface [1].

In this setting, we consider two protection schemes: (i) individual–individual and (ii) collective–individual. Quantum circuits of the state transfer experiments are shown in Figure 10. As the individual–individual scheme’s protecting unitaries, we take:(21)$$U={\tilde{U}}^{⊗2},\;\tilde{U}=XH,\;V={\tilde{V}}^{⊗2},\;\tilde{V}=HX.$$
In the case of the collective–individual protection scheme, we take"
(22)$$U={\left(X\right)}^{⊗2}B^{†}\left(\xi_{\mathrm{opt}}\right),\;V={\tilde{V}}^{⊗2}.$$

In our demonstrations, after preparing the input two-qubit state using the $U_{\mathrm{prep}}$ operator, we transfer the state of one qubit, via SWAP operations, through the environment initialized in the |0〉 state. Then, we reconstruct the whole two-qubit state $\rho_{\mathrm{out}}$ using the standard linear inversion technique, based on the symmetric informationally complete positive operator-valued measure (SIC-POVM) measurements, on each of the two qubits. The fidelity is calculated as $\langle \Psi_{\mathrm{in}}|\rho_{\mathrm{out}}|\Psi_{\mathrm{in}}\rangle$.

The SIC-POVM quantum state tomography circuit is shown in Figure 11. The tomography circuit utilizes an ancilla qubit for each target qubit. The corresponding circuit parameters realizing SIC-POVM measurements are given by
(23)$$\alpha =2\arccos\left(\sqrt{\frac{1}{2}+\frac{1}{2\sqrt{3}}}\right),\;\beta =\frac{\pi }{4}.$$

Physically, for tomography we use the same environment qubits as during state transfer since, after SWAP operations, the state of the mediated environment does not change. As we provide the real-world example, we optimize the transfer circuit to achieve maximally accessible fidelity. Therefore, we reduce the SWAP operations to two CNOT operations thanks to the knowledge of the initial state of the environment. The topology of the state transfer and optimized operations are shown in Figure 9.

The result of the optimization of the ξ parameter and the resulting input–output fidelities are shown in Figure 12. We note that as this is a practical example, the experimental data was obtained from transpiled circuits. We provide the results of several protocol runs corresponding to a different number of SWAP operations. One can observe a sizable increase in fidelity compared to the unprotected case.

It is worth mentioning that the considered approach for a state propagation protection can be used in distributed computing architectures, where two or more ‘stationary’ quantum processors are physically separated and connected by means of ‘flying’ qubits. The appearing decoherence channel then consists of imperfections related to interactions between stationary and flying qubits, as well a decoherence during the transmission of flying qubits through some medium, e.g., optical fibres or free space. Anyway, individual–individual and a collective–individual schemes can be used in this case, where collective operations are realized within a single quantum processor.

## 6. Conclusions

We have demonstrated the implementation of the scheme for suppressing decoherence in multi-qubit quantum systems according to a recent theoretical proposal [80]. The simulation of the scheme for suppressing decoherence in two- and four-qubit quantum systems was carried out for three main types of decoherence: particularly, depolarizing, dephasing, and amplitude damping, which were presented as quantum channels acting on each qubit of two- and four-qubit systems. As a result, the dependencies of the fidelity of the output quantum state on the strength of decoherence have been obtained. It has been shown that the most advantageous scheme of suppression, as expected, is the collective–collective scheme, which effectively suppresses decoherence in all cases. The realization of collective operators in the general case, however, requires an exponential number of elementary single- and two-qubit gates, which limits their applicability in realistic conditions.

We have shown the paradigmatic example of decoherence effects suppression in experiments with a cloud-accessible, five-qubit quantum processor ibmq_manila, where the strength of the decoherence is controlled by the delay. We have observed the results of increasing the fidelity value both in the case of two-qubit decoherence and in the case of four-qubit decoherence for all types of schemes. We also have demonstrated the real-world example of quantum state transfer with a cloud-accessible, seven-qubit quantum processor ibm_oslo. We have observed the increase in fidelity for optimized state transfer protocol up to 10%. We expect our findings to be useful for increasing the performance of current NISQ devices.

## Figures and Tables

**Figure 1 entropy-25-00067-f001:**
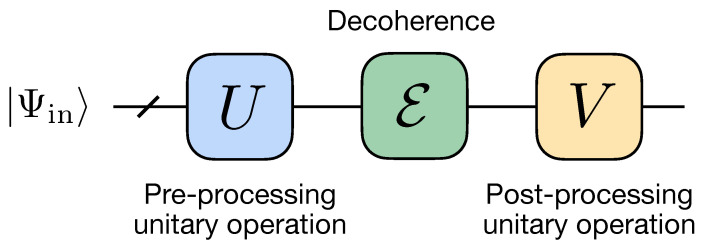
Quantum circuit of error suppression based on pre-processing and post-processing unitary operations.

**Figure 2 entropy-25-00067-f002:**
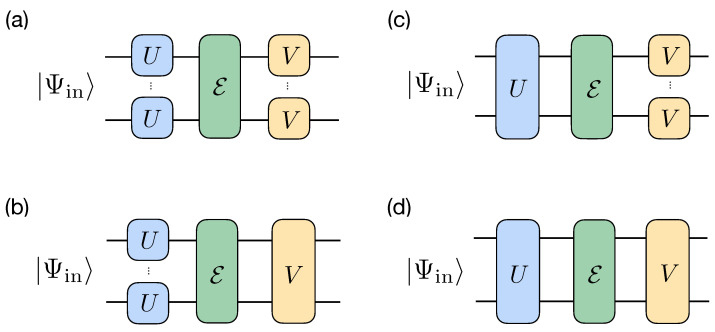
Schemes for protecting a pure state $|\Psi_{\mathrm{in}}\rangle$ from the decoherence channel E: individual–individual (**a**), individual–collective (**b**), collective–individual scheme (**c**), and collective–collective (**d**). Note that single-qubit *U* and *V* can be different for different qubits.

**Figure 3 entropy-25-00067-f003:**
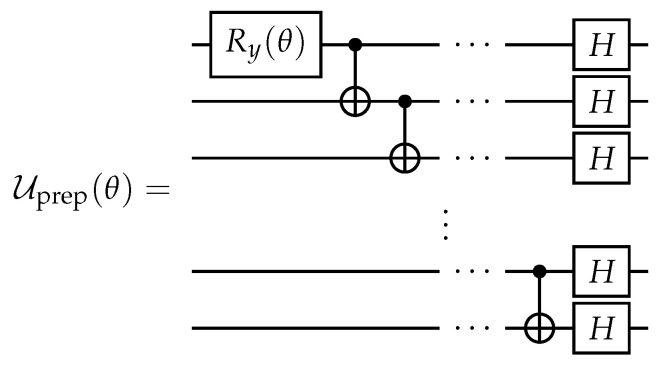
Quantum circuit for the unitary operator performing preparation of the *n*-qubit input state $|\Psi_{\mathrm{in}}\left(\theta \right)\rangle$.

**Figure 4 entropy-25-00067-f004:**
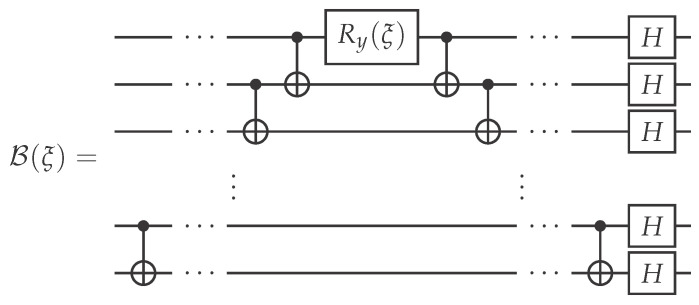
Quantum circuit of the collective operation B(ξ).

**Figure 5 entropy-25-00067-f005:**
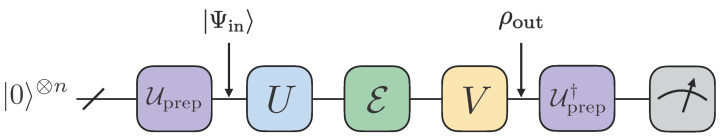
Circuit for the measurement of the resulting fidelity. Parameter θ for the $U_{\mathrm{prep}}$ is taken to be equal to 2π/3.

**Figure 6 entropy-25-00067-f006:**
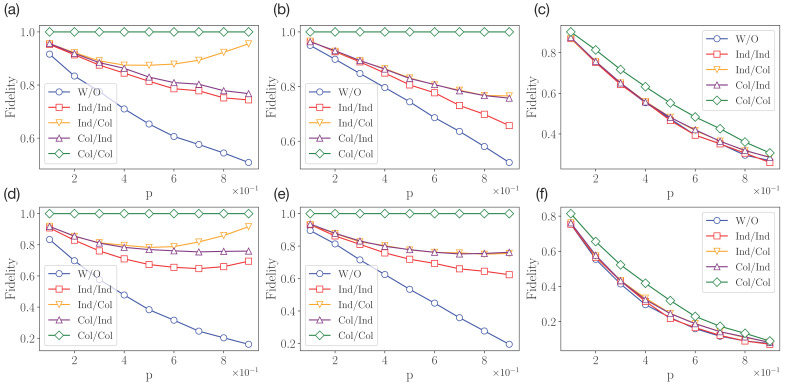
Simulation results for n=2 qubit (**a**–**c**) and n=4 qubit (**d**–**f**) cases. The amplitude damping (**a**,**d**), dephasing (**b**,**e**), and depolarizing (**c**,**f**) channels are considered.

**Figure 7 entropy-25-00067-f007:**
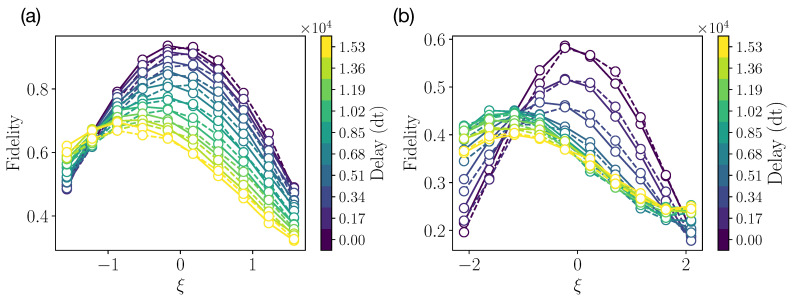
Experimental results of ξ angle calibration from IBMQ quantum processor for (**a**) two-qubit experiment and (**b**) four-qubit experiment. Solid and dashed lines stand for collective–individual and individual–collective schemes, respectively.

**Figure 8 entropy-25-00067-f008:**
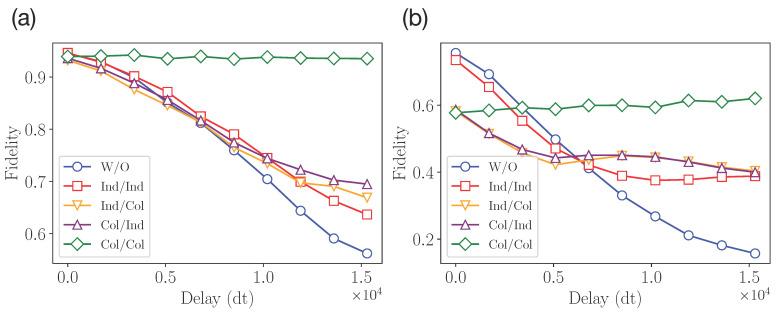
Experimental results from IBMQ quantum processor for (**a**) two-qubit experiment and (**b**) four-qubit experiment.

**Figure 9 entropy-25-00067-f009:**
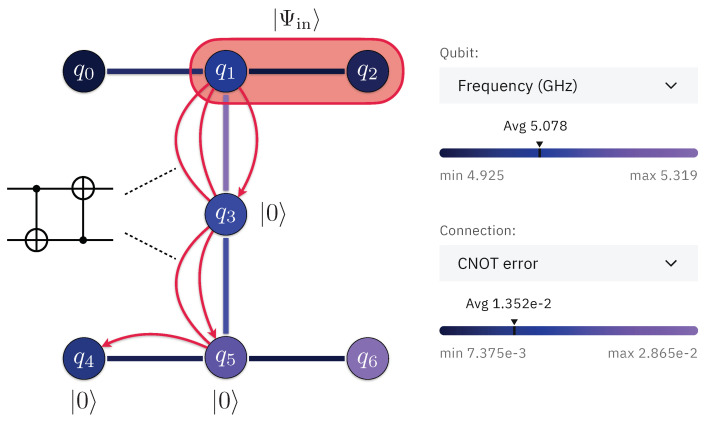
Scheme of the ibm_oslo quantum processor experiment (inset is from quantum-computing.ibm.com, accessed on 26 December 2022).

**Figure 10 entropy-25-00067-f010:**
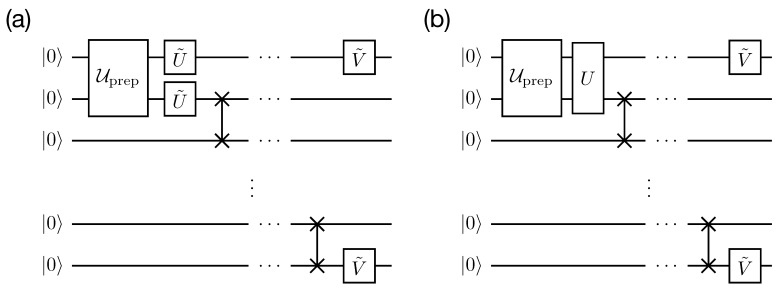
Quantum circuits of state transfer experiments with (**a**) individual–individual protection scheme and (**b**) collective–individual protection scheme.

**Figure 11 entropy-25-00067-f011:**
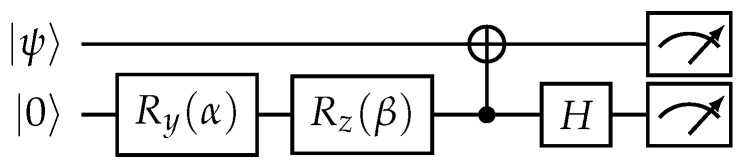
Quantum circuit for input state |ψ〉 tomography.

**Figure 12 entropy-25-00067-f012:**
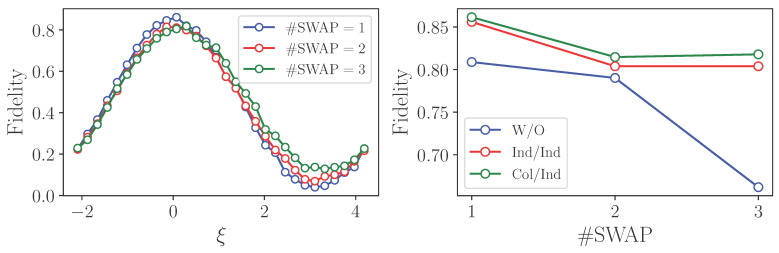
Experimental results of state transfer from IBM quantum processor.

## Data Availability

All data available upon the reasonable request.

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
