# Peer review of "Suppressing Decoherence in Quantum State Transfer with Unitary Operations"

_entropy, 2022, doi:10.3390/e25010067_

Round 1

Reviewer 1 Report

the manuscript proposes four schemes and specific quantum unitaries for pre- and post-processing of a quantum state subject to decoherence, with the purpose to suppress the effects of such decoherence.

The proposed approach is an extension of the approach proposed by the same authors in a previous work [39]. Overall, the approach seems to be sound and clearly exposed. However, the lack of references to other approaches (e.g., those from the huge literature on quantum error correction) is a major issue that should be addressed in a future revision of the manuscript.

I see also two other issues to be addressed:

1) Evaluate the overhead of the proposed schemes in terms of extra gates introduced into the quantum circuit. 

2) Explain more clearly how the proposed approach presented in Section 5 could be used in a distributed computing architecture where two or more QPUs are physically separated and connected by means of quantum channels (fibre optical channels, but also free space in some cases).

Author Response

Comment of the First Referee:

The manuscript proposes four schemes and specific quantum unitaries for pre- and post-processing of a quantum state subject to decoherence, with the purpose to suppress the effects of such decoherence.

The proposed approach is an extension of the approach proposed by the same authors in a previous work [39]. Overall, the approach seems to be sound and clearly exposed. However, the lack of references to other approaches (e.g., those from the huge literature on quantum error correction) is a major issue that should be addressed in a future revision of the manuscript.

I see also two other issues to be addressed:

1) Evaluate the overhead of the proposed schemes in terms of extra gates introduced into the quantum circuit. 

2) Explain more clearly how the proposed approach presented in Section 5 could be used in a distributed computing architecture where two or more QPUs are physically separated and connected by means of quantum channels (fibre optical channels, but also free space in some cases).

Response:

We are grateful to the Referee for the appreciation of our work! 

  1. The authors have added a discussion in the text after formula (4) about estimating the overheads of the proposed schemes in terms of extra gates introduced into the quantum circuit. A constructing collective operators yields an additional overhead in the number of gates and the corresponding circuit depth: Individual operators have a unit-depth and consist of no more than n single-qubit gates, while a transformation between two given pure n-qubit states, which is an aim of collective operators, generally requires an exponential in n number of single- and two-qubit gates or an exponential number of ancillary qubits for linear depth circuits. The authors also have added references to papers related to this. This corrections made by the authors are highlighted in blue color in the current version of the manuscript.

  1. The authors have added a discussion in the Sec.V. about distributed computing architecture. It is worth to mention that the considered approach for a state propagation protection can be used in distributed computing architectures, where two or more `stationary' quantum processors are physically separated and connected by means of `flying' qubits. The appearing decoherence channel then consists of imperfections related to interactions between stationary and flying qubits, as well a decoherence during transmission of flying qubits through some medium, e.g. optical fibres or free space.

Reviewer 2 Report

Dear Authors,

The paper describes experiments carried out on IBM Quantum’s simulators and real devices. The experiments pertain to the performance of unitaries specifically chosen for each of the presented three noise channels. The experiments collaborate theoretical findings in general.

I believe that the paper would benefit from the following adjustments:

1.       Adding a discussion on why the collective-collective scheme of protection does not perform well for depolarizing channels as shown in Figure 6 (c, f).

2.       There is quite a significant number of typos in the text. Some spotted ones are pointed out below. The authors are advised to revise the text in full to make corrections.

Line 43: replace “remain” with “remains”

Line 45: replace “stress on” with “stress”

Line 49: replace “protecting state” with “protected state”

Line 51: replace “aall” with “all”

Line 71: replace “can be either individual and collective” with “can be either individual or collective”

Line 72: replace “are relate to each other” with “are related to each other”

Line 88: replace “before proceed” with “before proceeding”

Line 91: replace “assumed to perfect” with “assumed to be perfect”

Line 93: replace “as the part of the qiskit package” with “as part of the qiskit package”

Line 113: replace “case individual-collective scheme” with “case of individual-collective scheme”

Line 120: replace “to three introduced types” with “to the three introduced types”

Line 127: replace “this behavior can explained” with “this behavior can be explained”

Line 129: replace “is thus is” with “is thus”

Line 130: A period is missing in front of “in the case”

Line 151: replace “with the enhance” with “with the enhancement”

Line 179: The sentence is ungrammatical. Please correct.

3.       As the special issue the paper was submitted to talks about quantum information in a wide scope, the introduction section should be amended to incorporate a general description of the field of quantum information to make it complete. I suggest to add a paragraph outlining the development in the field to include references to such topics as given below with some suggested references:

a.       quantum cryptography based on fundamentals of quantum mechanics:

[1] Do Ngoc Diep, Koji Nagata, Renata Wong. Continuous-variable quantum computing and its applications to cryptography. International Journal of Theoretical Physics 59(10): 3184-3188, 2020. Doi: 10.1007/s10773-020-04571-5

b.       quantum information for biology and medicine:

[1] Weng-Long Chang, Ju-Chin Chen, Wen-Yu Chung, Chun-Yuan Hsiao, Renata Wong, Athanasios V Vasilakos. Quantum speedup and mathematical solutions of implementing bio-molecular solutions for the independent set problem on IBM quantum computers. IEEE Transactions on NanoBioscience 20(3): 354-376, 2021. Doi: 10.1109/TNB.2021.3075733

[2] Renata Wong and Weng-Long Chang. Fast Quantum Algorithm for Protein Structure Prediction in Hydrophobic-Hydrophilic Model. Journal of Parallel and Distributed Computing 164: 178-190, 2022. Doi: 10.1016/j.jpdc.2022.03.011

[3] W. -L. Chang, J. -C. Chen, W. -Y. Chung, C. -Y. Hsiao, R. Wong and A. V. Vasilakos . Quantum Speedup for Inferring the Value of Each Bit of a Solution State in Unsorted Databases Using a Bio-Molecular Algorithm on IBM Quantum's Computers. IEEE Transactions on NanoBioscience 21(2): 286-293, 2022. Doi: 10.1109/TNB.2021.3130811

[4] Renata Wong and Weng-Long Chang. Quantum Speedup for Protein Structure Prediction. IEEE Transactions on NanoBioscience 20(3): 323-330, 2021, doi: 10.1109/TNB.2021.3065051.

c.       quantum machine learning:

[1] J. Biamonte, P. Wittek, N. Pancotti, P. Rebentrost, N. Wiebe, S. Lloyd. Quantum machine learning. Nature 549: 195-202, 2017. Doi: 10.1038/nature23474

Author Response

Dear Authors,

The paper describes experiments carried out on IBM Quantum’s simulators and real devices. The experiments pertain to the performance of unitaries specifically chosen for each of the presented three noise channels. The experiments collaborate theoretical findings in general.

I believe that the paper would benefit from the following adjustments:

  1.       Adding a discussion on why the collective-collective scheme of protection does not perform well for depolarizing channels as shown in Figure 6 (c, f).

  1.       There is quite a significant number of typos in the text. Some spotted ones are pointed out below. The authors are advised to revise the text in full to make corrections.

Line 43: replace “remain” with “remains”

Line 45: replace “stress on” with “stress”

Line 49: replace “protecting state” with “protected state”

Line 51: replace “aall” with “all”

Line 71: replace “can be either individual and collective” with “can be either individual or collective”

Line 72: replace “are relate to each other” with “are related to each other”

Line 88: replace “before proceed” with “before proceeding”

Line 91: replace “assumed to perfect” with “assumed to be perfect”

Line 93: replace “as the part of the qiskit package” with “as part of the qiskit package”

Line 113: replace “case individual-collective scheme” with “case of individual-collective scheme”

Line 120: replace “to three introduced types” with “to the three introduced types”

Line 127: replace “this behavior can explained” with “this behavior can be explained”

Line 129: replace “is thus is” with “is thus”

Line 130: A period is missing in front of “in the case”

Line 151: replace “with the enhance” with “with the enhancement”

Line 179: The sentence is ungrammatical. Please correct.

  1.       As the special issue the paper was submitted to talks about quantum information in a wide scope, the introduction section should be amended to incorporate a general description of the field of quantum information to make it complete. I suggest to add a paragraph outlining the development in the field to include references to such topics as given below with some suggested references:

  1.       quantum cryptography based on fundamentals of quantum mechanics:

[1] Do Ngoc Diep, Koji Nagata, Renata Wong. Continuous-variable quantum computing and its applications to cryptography. International Journal of Theoretical Physics 59(10): 3184-3188, 2020. Doi: 10.1007/s10773-020-04571-5

  1.       quantum information for biology and medicine:

[1] Weng-Long Chang, Ju-Chin Chen, Wen-Yu Chung, Chun-Yuan Hsiao, Renata Wong, Athanasios V Vasilakos. Quantum speedup and mathematical solutions of implementing bio-molecular solutions for the independent set problem on IBM quantum computers. IEEE Transactions on NanoBioscience 20(3): 354-376, 2021. Doi: 10.1109/TNB.2021.3075733

[2] Renata Wong and Weng-Long Chang. Fast Quantum Algorithm for Protein Structure Prediction in Hydrophobic-Hydrophilic Model. Journal of Parallel and Distributed Computing 164: 178-190, 2022. Doi: 10.1016/j.jpdc.2022.03.011

[3] W. -L. Chang, J. -C. Chen, W. -Y. Chung, C. -Y. Hsiao, R. Wong and A. V. Vasilakos . Quantum Speedup for Inferring the Value of Each Bit of a Solution State in Unsorted Databases Using a Bio-Molecular Algorithm on IBM Quantum's Computers. IEEE Transactions on NanoBioscience 21(2): 286-293, 2022. Doi: 10.1109/TNB.2021.3130811

[4] Renata Wong and Weng-Long Chang. Quantum Speedup for Protein Structure Prediction. IEEE Transactions on NanoBioscience 20(3): 323-330, 2021, doi: 10.1109/TNB.2021.3065051.

  1.       quantum machine learning:

[1] J. Biamonte, P. Wittek, N. Pancotti, P. Rebentrost, N. Wiebe, S. Lloyd. Quantum machine learning. Nature 549: 195-202, 2017. Doi: 10.1038/nature23474

Response:

We are grateful to the Referee for the appreciation of our work! 

  1. The authors have added a discussion in the text after formula (18), wich answer to a question why the collective-collective scheme of protection does not perform well for depolarizing channels as shown in Figure 6 (c, f). The reason for the this fact is that depolarizing channel always increases mixedness of input states, and there is no pure state that is preserved under the depolarizing. This corrections made by the authors are highlighted in blue color in the current version of the manuscript.

  1. The authors thank the Reviewer for careful reading of our manuscript and useful comments. According to all the recommendations, the authors completely revised the text in full to make corrections. All corrections made by the authors are highlighted in red color in the current version of the manuscript.

  1. The authors are grateful to the referee for pointing out on this references. Although authors are not able to cover the whole development on the field of quantum information since it is requires specific review, e.g. in arXiv:2203.17181 (Quantum computing at the quantum advantage threshold: a down-to-business review). The authors at the same time cited review papers on the field of quantum information and linked our results to its development.

With kind regards

The authors.

Round 2

Reviewer 1 Report

I am fine with most of the revisions, but I suggest two further improvements:

1) add references to other approaches (e.g., those from the huge literature on quantum error correction)

2) in the Conclusions, remark that collective operations, despite very effective, introduce an exponential overhead in the number of gates and ancilla qubits

Author Response

We thank the referee for the comments, which we fully agree on. We have added new references to the paper (see Introduction) and added a remark on the number of gates (first paragraph of Conclusion).

Reviewer 2 Report

As the authors have not responded to all comments, I suggest a major revision of the paper with the request that the authors address all points in the review. 

Entropy is a good journey. There are many readers in Entropy. Referee recommends several articles that are the top work in the different fields with quantum computers. Referee think after the reader read the revised version with my recommended articles, the reader can understand the newest researching work and finds their perhaps study direction. This will give the reader many help and interesting.

Author Response

We thank the referee for the comments. We have substantially improve the range of references in the new version of the manuscript.